# Opinion: Should high resolution differential mobility analyzers be used in mainstream aerosol studies?

Juan Fernandez de la Mora

Department of Mechanical Engineering, Yale University, 9 Hillhouse Avenue, New Haven, CT 06520-8286, USA

*Correspondence to*: Juan Fernandez de la Mora (juan.delamora@yale.edu)

**Abstract.** Differential Mobility Analyzers (DMAs) are widely used instruments to measure the size distributions of submicron aerosols. High Resolution DMAs (HRDMAs) are defined here as plain DMAs maintaining a steady flow over an unusually broad range of sheath gas flow rates $Q$. HRDMAs, first developed by Georg Reischl's group, have existed for a long time. However, they have not been widely adopted, except in the size range below 10 nm, often in new particle formation studies. Here we question the commonly held view that HRDMAs are necessarily complex, bulky and expensive machines, mainly of interest in exotic applications outside mainstream aerosol research. Rather, many studies central to aerosol research could be carried out with HRDMAs with considerable advantage in size range, resolution, sensitivity, and measurement speed. DMA manufacturers will hopefully take the challenge of developing commercial HRDMAs of complexity and cost comparable to those of today's commercial instruments, adapted for broad use by aerosol scientists, though with greatly improved flexibility and performance. Some of the technical challenges that still need to be overcome are discussed, such as the development of high flow condensation counter detectors, and the control of high sample and sheath gas flow rates.

## 1. Introduction

Flagan (1998) has described the long and complex evolution of electrical aerosol measurement methods, culminating in the modern development of the differential mobility analyzer (DMA) by Liu and Pui (1974) and Knutson and Whitby (1975). DMAs have subsequently become irreplaceable instruments, extensively used in studies involving submicron particles. Here we shall focus on high resolution DMAs (HRDMAs), defined as DMAs operating steadily at unusually high flow rate of sheath gas (HFDMA). One possible quantitative definition is

HRDMA=HFDMA=DMA operating steadily at $Q > 100$ L/min.

It is nevertheless preferable to define HRDMAs more qualitatively as DMAs *including special features enabling an extended Q range*. Likewise, we shall denote as *plain* DMAs those having no such special features.

A distinction must be made in this respect between subcritical, critical and supercritical DMAs, based on the fact that laminar flow in a tube, or in a channel between two plates (or two concentric tubes) typically becomes unstable at Reynolds numbers (Re) in the range of 2000 (depending on geometry). Supercritical DMAs may roughly be defined as those keeping the flow laminar well above Re=2000 by carefully laminarizing the sheath flow, and by also avoiding any slight discontinuity on the wetted walls. Critical DMAs could be similarly defined as those keeping the flow laminar up to Re=2000, but not much beyond. Subcritical DMAs would then be those including unusual features forcing flow instabilities well below Re=2000. This distinction is not irrelevant because laminar DMA operation at Re well above 100,000 has been demonstrated many times, while most commercial DMAs tend to operate considerably below Re=2000. Accordingly, because critical DMAs would represent a substantial improvement over the usual current situation, we will treat them as HRDMAs for the present discussion.

Only a minor fraction of the large bibliography on DMAs has centered on applications benefitting from a relatively high resolution, mostly focusing on particles smaller than 10 nm (Kangasluoma et al. 2020; Ozon et al. 2021). This has created an impression that the main utility of HRDMAs is in the low *nano* corner of aerosol research, rarely well resolved by more conventional DMAs. The question examined in this *opinion* is whether it would make sense to use future HRDMAs in situations presently handled by *plain* DMAs.  We anticipate that the answer will be YES, primarily due to considerable benefits in terms of size range, resolution and sensitivity.

At first sight it would appear that HRDMAs are unlikely to play a role in mainstream aerosol research for a variety of reasons. There is first the perception that HRDMAs are complex, heavy, and expensive machines, requiring large pumps, cooling systems, long diffusers, wide inlets, etc. For instance, Kangasluoma et al. (2017) note that *"However, these supercritical DMAs were mainly used in laboratory calibrations in previous studies because of their high flow rates of several hundred or thousand liters per minute and the corresponding high maintaining expenses"*. Similarly, we read in Cai et al. (2017) that *"However, these DMAs are designed to operate at high sheath flowrate (typically several hundreds to thousands lpm) rather than matching the flowrate required by the DEG-UCPC. In addition, large sheath flow recirculation system (including high volume filters and air mover) are required to operate those DMAs, making it troublesome for field measurement"*.

These remarks originate from groups well acquainted with the characteristics of HRDMAs through their pioneering studies on new particle formation. They should accordingly not be lightly dismissed. To the least, they suggest that the wide use of HRDMAs in aerosol research will require new developments. More fundamentally, many aerosol studies cover a vast size range, rarely including narrowly defined features demanding high resolution. Furthermore, if this extensive size range were probed at high resolution, it would apparently take a long time to do so, with each narrow size range individually examined containing too little signal. These theoretical reasons seem to be confirmed by the observation that the vast majority of DMA users have relied on long established commercial instruments operating at resolving powers typically of 10 or less. DMA resolution is defined here as the inverse of the relative full width at half maximum (*FWHM*) of the transfer function.

To guide the process of re-examining these various apparently persuasive notions against the wider use of HRDMAs, let us consider the main operating DMA parameters: the flow rates of polydisperse and monodisperse aerosol, $q$ (taken for simplicity to be identical), the flow rate $Q$ of sheath gas,  the classification voltage $V$ (Figure 1), and the mean electrical mobility $Z$ of the classified particle. For planar and axisymmetric instruments, these quantities are simply connected by equation (1) through a single parameter $k$, fixed by the instrument geometry:

$Z=kQ/V$.                                                                                                        (1)

The relative width of the transfer function, *FWHM,* also depends on $q$, $Q$, $V$ and $Z$, mainly through $q/Q$, and $V$ (Knutson and Whitby, 1975; Flagan, 1999)

$FWHM=q/Q$ for non-diffusive particles,                                                                (2)

$FWHM^2 \sim V/(k_B T)$ for diffusive particles when $q<<Q$,                                    (3)

where $k_B T$ is the thermal energy.

Typically, $k$ and $Z$ are externally imposed, with little room to maneuver (though $k$ may be scanned in DMAs of variable geometry: Bezantakos et al., 2016; Lee et al., 2020; Perez Lorenzo et al., 2020a). The control parameters most readily available to the user are accordingly $Q$ and $V$.

The voltage is in most DMAs widely variable over an extended range, from a few volts up to the maximum value prior to electrical breakdown, typically between 5 and 15 kV, depending on geometry. The advantage of this ample $V$ range is evident from Equation (1), as it enables spanning a comparable range in particle mobility. In contrast, $Q$ is most often varied over a much narrower range, from about 1 L/min to 40 L/min. Why this limited $Q$ range is at first sight puzzling, as, in view of (1), the accessible $Z$ range depends as much on $V$ as on $Q$. One's puzzlement increases further when noting that the

resolving power also depends as strongly on $V$ as on $Q$, since both are coupled through (1). A DMA with a narrow $Q$ range necessarily offers far less operational flexibility than another with a wider $Q$ range. We shall see that a limited $Q$ range implies a limited performance not only in terms of size range and resolving power, but also in sensitivity, and speed of response.

## 2. DMA characteristics

**2.1 Size range, resolving power, sensitivity and optimal scan.**

According to Eq. (1), a mobility spectrum may be acquired as well by scanning over $Q$ as over $V$ (with technical difficulties to be discussed in Sect 3.3). Suppose one needs to cover the mobility range $Z_{max} \leq Z \leq Z_{min}$. Given that the diffusion-limited resolving power increases as $V^{1/2}$, [Equation (3)] the optimal scan would start with the largest particle, $Z_{min}=kQ_{min}/V_{max}$, and evolve over increasing $Q$ values, with $V$

fixed at $V_{max}$. An additional advantage of this $Q$ scan over conventional $V$ scans (besides its minimization of diffusive broadening) is that one may increase $q$ proportionally to $Q$ to improve the sensitivity without resolution loss. Once the limit $Q=Q_{max}$ is reached, the mobility spectrum may be extended by scanning over decreasing $V$ at fixed $Q=Q_{max}$. The range $Z_{max}/Z_{min}$ in this scan is $Q_{max}V_{max}/(Q_{min}V_{min})$. This range applies equally to *plain* and high resolution DMAs. The difference is

that the former may typically vary $Q$ from 2 to 25 L/min, while the later may reach well beyond 1000 L/min. Some level of $Q$ extension (not necessarily up to thousands of L/min) is well known to be essential to analyze 1 nm particles without excessive diffusive broadening.

**2.2 Response time.**

Another important advantage of HRDMAs relates to permissible scan speed (Fernandez de la Mora et

al., 2017a). High $Q$ implies a reduced response time, but fast measurements do not strictly require high flow rates. High resolution is even more useful, as it implies that particles of a given $Z$ are classified over a narrow range of $V/Q$ values. As a result, they exit the DMA at almost the same time during a mobility scan. If the scan is fast, the spectrum will tend to be distorted both in *plain* DMAs and HRDMAs. The reason is that, at the time the DMA voltage is $V(t)$, the detector senses a signal $I(t-\Delta t)$

corresponding to the DMA voltage (or flow rate) applied a certain time delay $\Delta t$ earlier. Nevertheless, the mobility spectra in HRDMAs are undistorted (other than through this uniform time shift), whence

the original size spectrum may be recovered via a simple translation of the voltage in time. In contrast, in *plain* DMAs, particles of a given mobility are classified over a wider range of $V/Q$ (scan times), resulting in peak shape distortions that are not simple time delays. Recovery of the undistorted signal is in this case far from trivial, forcing considerably slower scans than achievable in HRDMAs. Greater speed of measurement is a characteristic of universal interest in all situations where the signal is strong enough. This has led to the recent commercial development of a variety of condensation particle counter detectors (CPCs) with relatively fast responses. Why would DMA manufacturers not take advantage of this notable advance by developing HRDMAs generating undistorted size spectra with response times comparable or better than those of contemporary fast CPCs?

## 2.3 Presumed irrelevance of HRDMAs.

Let us provisionally assume that high resolution is not necessary in most aerosol studies spanning a wide size range, on the grounds that no narrow features exist in typical size spectra. Let us further accept that not enough signal is available over narrow size regions, and that most aerosol studies do not need to cover the size range below 5-10 nm. A HRDMA would still be substantially more useful than a plain DMA. Indeed, thanks to its broad $Q$ range, its resolving power and size range can be controlled far more widely than in plain DMAs. Complete freedom to run HRDMAs at limited resolution, when desired, naturally hinges on the ability to increase the aerosol sample flow $q$ proportionally to $Q$. This would simultaneously greatly increase sensitivity, largely controlled by $q$. Increasing $q$ is straightforward with existing electrometer detectors. However, Susanne V. Hering (private communication) has perceptively pointed out that high flow CPCs do not presently exist. In order to flexibly increase the sensitivity and control the resolution of HFDMAs, one would need to develop them together with high flow CPCs. This double challenge may seem non-trivial, but success in it would have a large impact on aerosol research.

We have so far considered studies of particles larger than 5-10 nm at moderate resolving power. There are nevertheless situations of clear interest calling for the classification and detection of ultrafine particles with resolving powers higher than available with plain DMAs. An example is the formation of new atmospheric particles and their subsequent growth (Kong, et al. 2021; Kangasluoma et al. 2020; Ozon et al. 2021). If the nucleation event is brief and the growth period long, as often happens, the new particles will tend to have narrow size distributions. A plain DMA will not determine as fast, sensitively, and accurately either the particle size or the growth rate. It will not even see a narrow size distribution if it exist. It is not widely realized that the sensitivity with which one can detect narrowly defined size distributions increases rather than decreasing at increasing resolving power. We observe this all the time in the study of viruses, with peaks either narrow and isolated, or wider and partially buried within a large background, at high and low flow rates, respectively (See Figures 2 and 3 of Fernandez de la Mora et al. 2021). The reason is that the signal is concentrated over a narrow mobility range, while the noise is spread continuously over a much wider domain. Capturing the whole signal over a narrow range of mobilities therefore reduces greatly the noise but not the signal. In contrast, capturing the particles over a wider mobility window does not increase the signal, yet augments the noise. The ideal resolving power from the strict point of view of signal/noise is accordingly dependent on circumstances. A flexible instrument where the resolving power and the sensitivity may be tuned as

required by these circumstances is evidently better suited for this and other comparably demanding applications. Likewise, a flexible single instrument able to cover the nanorange (Perez-Lorenzo et al. 2021) as well as 200-300 nm particles (Fernandez de la Mora et al. 2023; Fernandez de la Mora and Papanu, 2023) is far more convenient than investing in two different instruments, one for each of these two ranges. Not to mention the issue of matching in a single size spectrum the outputs of two devices with different characteristics.

## 2.4 Widening the $Q$ range: is it so hard?

The simple criterion adopted here based on flow rate of sheath gas appears to facilitate the classification of the numerous commercial DMAs in the market. Nevertheless, even when the manufacturer indicates a modest maximal or operating $Q$, the instrument may in fact accept much larger flows. For instance, in their first detailed description of Reischl's short 1/40 DMA (nominally classifying particles from 1 to 40 nm in diameter, but really going up to 150 nm), Winklmayr et al. (1991) indicated an operational flow rate $Q$=28 L/min. However, Rosell et al. (1996) found that they could draw $Q$ values beyond 300 L/min with hardly any changes, and even beyond 800 L/min by adding two more exhaust lines to the single original exhaust in the sheath gas manyfold (de Juan et al. 1998). Furthermore, their shorter version of this 1/40 DMA maintained the flow laminar over most of this considerable $Q$ range. Reischl's short 1/40 DMA therefore qualifies *de facto* as the first HRDMA.

The first surprising feature we discovered in our study of TSI's 3071 DMA was that the Reynolds number in its annular classification region was only a few hundreds at the highest recommended flow rates of 20-40 L/min, at which some flow instability was already present. These are the flow rates typical of other commercial DMAs, to which this puzzling situation may also apply. One would certainly expect a serious deterioration of the performance at increasing Reynolds numbers due to turbulent transition. But this transition is not supposed to take place until Reynolds numbers (Re) well above 1000. And even above that critical value, turbulent transition takes a considerable length to develop, especially when the inlet flow has been carefully laminarized, or when the working section is slightly converging. As theoretically expected, the unnaturally precocious flow instability in TSI's 3071 DMA could be removed by avoiding two types of aerodynamic problems: (i) steps following immediately after the inner and outer radii of the laminarization screen, and (ii) unstable regions with decelerating boundary layers in the sheath gas inlet (Eichler et al. 1998; Fernandez de la Mora et al. 2017b). Once these aerodynamic extravagances were cleared, widening a few downstream features offering excessive flow resistance enabled reaching $Q$=100 L/min without any signs of flow instability. However, although the resolution increased considerably, it remained well below the theoretical value dictated by Brownian diffusion and by the finite value of $q/Q$. This latter result suggested that flow perturbations injected at the laminarization screen are a real problem in strictly cylindrical DMAs. Indeed, since the flow cross section is the same in the laminarization screens and in the classification region, the constriction created by the screens accelerates locally the flow into a multitude of jets, whose decay is by no means immediate. Accordingly, the famous 3071 DMA and various generations of successors still limited to flow rates below 25-40 L/min at TSI and elsewhere, must presently be classified as *plain* DMAs. Nevertheless, based on the precedents just discussed, some among these *plain* instruments could possibly approach and even reach the HRDMA category with limited changes.

## 2.5 Reischl's inlet trumpet and its minimization.

The screen problem just described for strictly cylindrical DMAs must have been known to Georg Reischl when he developed the first HRDMA featuring a trumpet-shaped sheath flow inlet, such that the cross section of the laminarization screens was substantially wider than that in the analyzing region. The trumpet included in Reischl's 10/40 DMA deserves some comment, as it was not discussed in any of Reischl's published articles, and does not even appear in the *schematic* in Figure 1 of Winklmayr et al. (1991). This trumpet-less published *schematic* was apparently used by others in the development of various clones circulated in European laboratories. Some at least among these copies did not enjoy the extended $Q$ characteristics of the original design, and were the cause of some confusion. For instance, Rosell et al. (1996) include the following footnote. *"... a preliminary test of a shortened Reischl DMA was performed .... However, for reasons never fully understood, that short DMA did neither yield the predicted resolution, nor did it operate properly at flow rates in the range of 80 L/min or above."* The mystery noted was simply the absence of that inlet trumpet in the cloned prototype used, as clarified years later in a private conversation with George Reischl (who stressed the distinction between a *schematic* and a *drawing*, as further clarified in the historically relevant discussions to this opinion by J. Rosell and Gerhard Steiner). Fortunately, the successful study by Rosell et al. (1996) had the benefit of Reischl's original drawings, including the inlet trumpet, liberally shared by their inventor with colleagues who requested them. Therefore, this inlet contraction must be taken to be an essential element in HRDMAs, at least until an alternative approach is demonstrated. An inlet trumpet was certainly part of all the successful DMAs developed at Yale. It is featured not only in prototypes and fabrication drawings of Reischl's 1/40 DMA, but also in publications describing his later high-$Q$ designs (Steiner et al., 2010; Keck et al. 2008), some of which were commercialized by Grimm Aerosol Technik GmbH. These more recent Reischl DMAs have been successfully tested at relatively high flow rates and do undoubtedly qualify as HRDMAs. The one Reischl model possibly belonging to the *plain category* is the long 10/1000 DMA, shown schematically in Figure 2 of Winklmayr et al. (1991). This DMA features a widening (destabilizing) rather than a converging section following the laminarization region, and operates nominally at $Q$=12 L/min. If the published *schematic* is faithful to the actual design, it is improbable that the 10/1000 DMA would have sustained a steady sheath flow at substantially higher $Qs$.

On the assumption that an inlet trumpet is essential, an important practical issue is to determine the smallest cross section ratio $A_s/A_w$ between the open screen area and the working section required by a HRDMA. I am not aware of any systematic study aimed at minimizing $A_s/A_w$, though our experience with TSI's 3071 DMA suggests that this area ratio must exceed unity. This does not necessarily mean that HRDMAs must be heavy and bulky. Three examples at least of hand-held HRDMAs have been described: Reischl's 10/40 (Winklmayr et al., 1991), the Half-Mini (Fernandez de la Mora, 2017) and the earliest version of the Perez DMA family (Perez-Lorenzo et al., 2020b). Worthy of note is the fact that Martinez-Lozano et al. (2006) have achieved resolving powers of 50 with tetraheptyl-ammonium ions in a peculiarly shaped (isopotential) DMA where the area ratio between the laminarization screen and the sampling location of classified particles was only 1.27. The high performance of their analyzer persisted up to the maximal flow rate tested of 2300 L/min! In a later study with an improved geometry,

Martinez-Lozano and Labowsky (2009) achieved a resolving power of 75 with the tetraheptylammonium ion.

## 2.6 Is the sheath gas circuit really so complex?

We now return to the issue of other heavy and bulky elements characteristic of a number of previously used HRDMAs. Please, note that the exotic applications of DMAs we have pursued at Yale, are not the same thing as creating a substantially improved instrument for broad use by aerosol scientists. As an example of the potential simplicity of the required system, I note that the large vacuum cleaner blowers consuming 1 kW of power we have often used to drive the sheath gas have tended to be large

inexpensive and relatively inefficient devices, both aerodynamically and electrically. This means that these pumps inject a lot of waste heat into the circulating gas, which must be removed by a relatively large heat exchanger. However, the recent development of battery operated portable vacuum cleaners has resulted in small blowers with fairly high aerodynamic and electrical efficiencies. Pérez-Lorenzo at al. (2017) have described one such commercial pump driving 200 L/min of sheath gas through a

HRDMA of relatively narrow cross section (inner and outer radii of 4 and 7 mm), while consuming only 12.5 Watt. Besides their small dimensions and weight, these efficient blowers heat minimally the recirculating gas, making the usual bulky heat exchanger unnecessary.

## 2.7 The role of geometry

Early DMAs were long (small $k$) to favor the classification of large particles. Kousaka et al. (1986)

demonstrated the interest of short DMAs (larger $k$) to diminish diffusion broadening of ultrafine particles. Geometry hence enables improving either $Z_{max}$ or $Z_{min}$ in plain DMAs by substantial factors. There is nevertheless a limit on how small a particle may be analyzed with fair resolving power by increasing $k$ (Rosell et al, 1996). As a result, two or three DMAs of different lengths are commonly offered commercially to span a wider size range than coverable by just one DMA. On the other hand, a

single HFDMA may sweep in a single scan from 1.5 nm to 300 nm, with a resolution in excess of 10. This is the natural consequence of Equation (1), where a wide change in $Q$ is equivalent to a wide change in $k$.
Our prior discussion has considered mainly traditional DMA geometries, involving cylindrical or slightly converging DMAs. The reason for this narrow scope is that these *axial flow* configurations are

270 the only ones where relatively high resolution and flow rate have been demonstrated to date. Flagan and his students have developed and demonstrated the advantages of so-called radial flow DMAs (Zhang et al. 1995). Radial and axial DMAs may be designed in a rich range of configurations far beyond what has been tested to date. Those so far explored generally have a flow field more or less perpendicular to the electric field. Diffusive broadening then arises mainly orthogonally to the fluid streamlines,

resulting in a resolving power scaling as $V^{1/2}$ [Equation (3)]. It then appears with considerable generality that the optimal resolving power at given $Q$ is reached at a certain optimal DMA length and cannot be further increased by geometrical manipulations (Fernandez de la Mora, 2002). Unless some unsuspected scheme is discovered, achieving high resolving power with ultrafine particles in these geometries necessarily requires relatively high flow rates. However, the situation is far more favorable when the

electric and the flow field are approximately opposed to each other, in which case the resolving power scales as $V$ instead of $V^{1/2}$. This may be theoretically demonstrated in a one-directional flow, for instance created between two porous plates held at different potentials. There is in principle no need of sheath gas, though the classification is not differential but cumulative, and a means to avoid complete loss of the particles through the porous medium is required. Differential separation can nevertheless be

achieved by combining opposing axial electric and flow fields with smaller lateral fields, as first demonstrated by the *Drift-DMA* configuration of Loscertales (1998). This most original proposal remained purely conceptual until Tammet (2011) implemented and tested it based on an inclined grid. The Opposed Migration Aerosol Classifier of Flagan (2004) adopted the one-dimensional geometry with two planar porous surfaces passing sheath gas rather than the aerosol (which moved laterally).

Neither of these configurations has yet demonstrated high resolving power, but their clear conceptual advantages suggest that it should be possible to create differential or cumulative mobility analyzers achieving high resolving power without requiring unusually large flow rates. Labowsky's isopotential DMAs (Martinez Lozano et al. 2006, 2009) are interesting cases where the flow and electric fields in the vicinity of the axis are opposed to each other, creating a stagnation point for the particle trajectories.

This stagnation may viewed as locally analogous to what happens globally in the strictly one dimensional uniform opposing field configuration. The isopotential DMA has demonstrated high resolving power, but not yet at modest flow rates.

**2.8 The use of high flow DMAs in monodisperse particle production**

The first anonymous referee has noted that further discussion of the aerosol sample flow rate is of clear broad interest. This is nicely illustrated by the pioneering work of Hontañón and Kruis (2009), whose study achieved the classification of substantial flow rates of monodisperse nanoparticles through the development of a DMA of heroic dimensions. Their goal was different from that of the present Opinion. Nevertheless, the referee's remark brings the important point that a small hand-held DMA capable of

high flow rate of sheath gas should also be able to handle high monodisperse aerosol flow rates, opening up another conceivable application of the DMA development I am trying to encourage. This interesting application would perhaps require somewhat different geometries (wider inlet and outlet slits, etc.). The main lesson following from the referee's insightful remark is that the flow rates may be increased by expanding the DMA dimensions while still running at moderate Reynolds numbers, as in

the critical DMA developed by Hontañón and Kruis (2009). The same goal may be reached by maintaining conventional moderate dimensions while increasing the Reynolds number well above 2000. This situation is not incompatible with laminar operation in properly designed DMAs, where steady flow has been demonstrated many times with Reynolds numbers well beyond 100,000.

**3. Technical problems requiring further developments**
**3.1 Detection.**
We have already mentioned Hering's key point that high flow CPCs do not presently exist and would need to be developed to exploit more fully the high sensitivity potential of HFDMAs. A recent study by Stolzenburg et al. (2023) has already operated a high flow Reischl DMA with a CPC drawing up to 2.5

L/min and demonstrated drastic sensitivity improvements in new particle formation research.

The discussions from Gerhard Steiner and one anonymous referee have rightly noted the possible advantages of electrical detectors, whose flow rates can be increased with far greater flexibility than in CPCs. Many readers may be skeptical about electrical detectors in atmospheric measurements, given that the typical electrometers used in aerosol research have noise levels of about 1-10 fA and response times of 1 s or more. They then require over $10^3$ elementary charges/s (or particles/s). Nevertheless, there is ample room for improving the response time and the sensitivity of most commercial aerosol electrometers. We recently reported the incorporation into a collecting filter of an operational amplifier circuit developed by Heinz Burtscher and his colleagues, which achieved a response time below 100 ms and a noise level of about 0.1 fA (Fernandez de la Mora et al. 2017). Nevertheless, this improvement is still modest compared to what could be achieved in practice. For instance, CCD cameras typically convert individual photons into elementary charges, and then measure their current. Such rapidly evolving charge detectors could therefore be applied to aerosol sensing. That the single particle sensitivity of CPCs is in principle extendible to electrical measurements has been clear for some time. For instance, Ma et. al. (2017) state that *"… the 1.1 μm pixel-pitch device achieves 0.21e− rms average read noise with average dark count rate per pixel less than 0.2e−/s, and 1040 fps readout rate"*. Notice that the outstanding sensitivity of their CMOS-based photon-counting image sensor applies not just to a single detector, but to arrays of many detectors ordered in two dimensions. This means that DMAs with such detector arrays would not need to scan over the voltage to obtain a size spectrum. The whole spectrum would be recorded almost instantly in a multitude of detectors distributed along the inner electrode. This possibility has already been demonstrated (Perez-Lorenzo et al. 2020) in a planar DMA with 100 operational amplifiers, though not yet with single charge sensitivity. This work shows that the insulating steps separating the various metallic collectors do not cause turbulent transition. A commercial linear detector array based on CCD technology with a noise level of 0.5 fA has existed for some time (https://www.jas-sg.com/ids-2030-charged-particle-detector.html). It is 51 mm long, has 2126 active pixels each 21 μm wide and 1500 μm high. It has been used in mass spectrometry based on instruments that disperse the ions in space. As far as we know, it has not been tested for mobility measurements, so, whether or not the collector roughness would cause turbulent transition is unknown. An array of electrical detectors with single charge sensitivity would be clearly superior to a DMA connected to a single CPC by a large factor equal to the number of mobility channels used. Note that the exceptional low noise achieved by Fossum's group (Ma et al. 2017) relies on the small area of each pixel. A typical situation with 1 million pixels, each 1.1 micron in length would appear at first sight to divide the signal into too many mobility channels, each receiving too little signal to be measurable in typical aerosol applications. Nevertheless, if the claimed ability of achieving single photon (electron) counting is correct, the signal would not be diluted over a wide mobility range, but assigned to precise values of the mobility. High mobility resolution would not imply in this case reduced sensitivity. One could certainly add the counts from as many pixels as desired by post-processing, without loss of the available high resolution information. Of course, the fact that such developments are in principle possible does not necessarily mean that they will be applicable soon to aerosol research.

The point of the first referee that the high sample flow rates manageable by electrometers may sometimes achieve greater sensitivities than existing CPCs is indeed nicely illustrated in various instruments developed by Tammet and his colleagues at Tartu (See for instance Asmi et al. 2009). It would be difficult to match the vast sample flow rates of these specialized instruments in a DMA

geometry. Nevertheless, it is conceivable that new instruments with resolving powers and sample flow rates intermediate between those of the Tartu analyzers and high flow DMAs could be developed by imaginative researchers.

## 3.2 HRDMA Characterization

Until recently, most existing HRDMAs had been characterized with respect to resolution at relatively high $Q/q$, but not under other flow rate conditions more common in many aerosol studies. Information on aerosol losses was also lacking, complicating the inversion of measured mobility spectra. This problem has fortunately been redressed in the recent study of Kangasluoma et al. (2018) for the special case of the Half-Mini DMA

## 3.3 Sheath flow rate control

Anonymous referee 2 brings up several important issues relating to the accurate control of the sheath gas. The proposed sweeping over $Q$ rather than $V$ requires a continuous way to measure $Q$, which cannot be achieved with the same precision and ease as a voltage measurement. There is also the issue of the much slower response time of mechanical pumps versus high voltage sources. Nevertheless, an accurate mobility scale requires a precise determination of $Q$, irrespective of whether one sweeps over $Q$ or over $V$. Whatever flowmeter is used, the ambiguity in the mobility scale will typically be determined by the flowmeter error, which will not be necessarily greater in a $Q$-sweep than in a $V$-sweep. The main difference is that in a $V$-sweep the $Z$ scale will be off by a fixed unknown factor, versus a variable equally unknown factor in a $Q$-sweep. Assuming that the relative error in the flow rate is not greater at high $Q$ than at low $Q$, it would be of little consolation to know that this error is constant. And this slight advantage will be lost in a conventional sweep if one decides not to be absolutely limited by operating always at fixed $Q$, irrespective of the size range of interest. These considerations naturally do not remove the need to measure $Q$ as precisely as possible over the whole range of the $Q$-sweep, which will evidently require a certain development. We are familiar with commercial flowmeters claiming 2% error in $Q$, and reproducibility within 1%, though only covering the limited range 0-300 L/min. Venturi flowmeters, for instance, can go substantially higher, with comparable precision. These various flowmeters have response times much faster than the pump (limited by the inertia of the rotor), being capable of determining the actual flow rate at any instant during a sweep. The instantaneous pump frequency is also readily measured, and is an alternative marker of the flow rate. There should accordingly be no difficulty in carrying out minute-long full $Q$ and $V$ scans. The possibility to achieve wide $V$-scans in a few seconds, previously demonstrated with electrometers (Fernandez de la Mora et al. 2017b), would be far more problematic in a $Q$-sweep.

There are, however, other potential difficulties associated to a $Q$-scan that may not be so simple to handle, or that could limit the scan speed. One of them is that the working pressure in a typical closed sheath flow circuit may depend on the pump speed, while the response time for this pressure depends on how the system pressure is set relative to the external world. No experience is available at present to assess this issue.

The ability to control the chemical composition of the sheath gas is relatively limited, as it is unpractical to dry a large flow rate of sheath gas. The most common way of operating a HRDMA is with the sheath gas in closed circuit, such that, after an initial transient, its composition matches that of the entering

aerosol. Zinola et al. (2019) have operated a high temperature Half-Mini DMA that sampled hot automobile exhaust gases at ambient pressure. To avoid vapor condensation from the exhaust gases they did run the sheath gas (and the DMA) at 200º C. This was achieved by taking filtered ambient gas into the pump, blowing it into a heater, then into the DMA sheath gas inlet, and finally exhausting the excess gas back to the atmosphere. The temperature was thus controlled, but not the humidity. Note that the pressure at the polydisperse aerosol inlet slit in the Half-Mini DMA is close to the pressure in the analyzing region, which is below the pressure at the entry of the sheath gas because the flow moves at relatively high speed (Venturi effect). This pressure is also below the sheath outlet pressure because the DMA is provided with a diffuser. Accordingly, given a suitable diffuser, it is possible to suck ambient aerosol when the sheath gas is exhausting to the atmosphere. On the other hand, none of the HRDMAs built to date to classify particles as large as 60 nm or more has incorporated a diffuser.

In the two previously tested modes, with open and closed sheath gas circuits, the only available compositions were either that of the ambient air or that of the aerosol. If one needs a sheath gas drier than the aerosol, one option is to dry the aerosol and operate in close circuit. Another option is to recirculate the sheath flow through a drying medium. It would be far simpler to remove the little humidity brough by the aerosol into the closed circuit than to thoroughly dry a large flow of ambient air. Other humidity controls are feasible. For instance, if the polydisperse aerosol flow is 3 L/min, and one introduces 27 L/min of dry gas into the closed circuit and (after thorough mixing of these two inputs) draws another 27 L/min out of the circuit, the sheath gas will reach an equilibrium humidity $1/10^{th}$ that of the entering aerosol.

## 4. Conclusions

I hope the general considerations and concrete examples provided here will help diffuse the notion that HFDMAs are unsuited for general aerosol studies, perhaps also stimulating their commercial development together with that of fast high-flow CPCs. The benefits to aerosol research would be very worth the effort.

**Competing interests**

The author is involved in the development and commercialization of several HRDMAS

**Author contribution**: The full opinion is the sole responsibility of the single author.

**Acknowlegments:**

I am much indebted to Susanne V. Hering and Michel Attoui for their insightful remarks on a first draft of this manuscript, and to the editors of Aerosol Research for their invitation to submit this study. Funding from AFOSR grant FA9550-22-1-0097 is gratefully acknowledged.

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

## Figures and captions

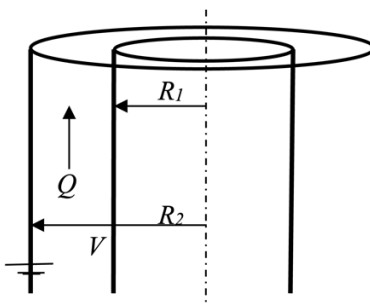

535 **Figure 1: Sketch of a cylindrical DMA with grounded outer electrode of radius $R_2$ and inner electrode of radius $R_1$ held at potential $V$, with a volumetric flow rate $Q$ of sheath gas moving axially and symmetrically between both electrodes**