# Peer review of "Opinion: Should high resolution differential mobility analyzers be used in mainstream aerosol studies?"

_Aerosol Research, 2023_

## Author Response (AR2)

**Response to reviewers**

I thank the author of **RC1**: 'Comment on ar-2023-7', Anonymous Referee #1, 29 Oct 2023 for these excellent remarks and bibliographical references. Most of them will be included in the revised

5 manuscript. Notice however that the work of Steiner, Attoui, Wimmer& Reischl was already cited.

I agree that further discussion of the aerosol sample flow rate is of clear broad interest, and that the reference to the work of Hontañón and Kruis is most appropriate. Their study achieved the classification of substantial flow rates of monodisperse nanoparticles through the development of a DMA of heroic dimensions. Their goal was substantially different from that of the Opinion under discussion.

- 10 Nevertheless, the referee's remark brings the important point that a small hand-held DMA capable of high flow rate of sheath gas should also be able to handle high monodisperse aerosol flow rates, opening up another conceivable application of the DMA development I was trying to encourage. This interesting application would perhaps require somewhat different geometries (wider inlet and outlet slits, etc.). The main lesson following from the referee's insightful remark is that the flow rates may be
- 15 increased by expanding the DMA dimensions while still running at moderate Reynolds numbers, as done by Hontañón and Kruis. The same goal may be reached by maintaining conventional moderate dimensions while increasing the Reynolds number well above 2000. This situation is not incompatible with laminar operation in properly designed DMAs, where steady flow has been demonstrated many times with Reynolds numbers well beyond 100,000.
- 20 The referee reminds us also of the potential interest of using an electrometer as a detector. The prior comment by Dr. Gerhard Steiner has also noted this fact. In my response to Gerhard's suggestion I have expanded further the discussion. The referee's point that the high sample flow rates manageable by electrometers may sometimes achieve greater sensitivities than existing CPCs is indeed nicely illustrated in various instruments developed by Tammet and his colleagues at Tartu (such as the
- 25 referee's first reference to Asmi et al.). It would be difficult to match the vast sample flow rates of these specialized instruments in a DMA geometry. Nevertheless, it is clear that this opinion should include a related discussion aimed at stimulating the creation of new instruments with resolving powers and sample flow rates intermediate between those of the Tartu analyzers and high flow DMAs.
- The referee's reference to Brunelli et al. involves an instrument operating at sheath gas flow rates and resolving powers comparable to those of conventional DMAs. I fully agree with the proposition that the creative geometrical innovations introduced in electrostatic classifiers (and their possible future variants) by Professor Flagan and his students offer great promise for high resolution classification, perhaps even without a need for high flow rates. Nevertheless, it seems to me that this important area of future developments is already discussed in my section 2.7, on the role of geometry.
- 35 Thank you again

I thank Joan Rosell-Llompart for his historical testimony. I should merely add that his recollections are 40 far more valuable than mine. First, because my memory leaves much to be desired. Also, and more importantly, because Joan was completely in charge of our group's program to make and test a shorter version of the Reischl DMA. I should also note that this program was started at Joan's initiative, when I reported that the tests of a shortened Reischl DMA at Duisburg by Andreas Schmidt-Ott and myself had not increased the resolving power as theoretically expected.

45

I am most grateful to Gerhard Steiner for his remarks. His endorsement of the program I was proposing is especially welcome given his position of responsibility for instrument development at Grimm.

It is most unfortunate that Georg Reischl, like Stradivarius, passed away without elaborating on the 50 design criteria used in his DMAs. Gerhard's testimony relating to Georg's views on the importance of accelerating the sheath gas past the laminarization screens is therefore a valuable confirmation of a principle observable in the genuine Reischl prototypes, but not in his published drawings and in many clones circulating in European labs. I have nevertheless a slight disagreement with Gerhard's account. In my own recollection, my only conversation with Georg was not by telephone, but in person, during an

55 European aerosol conference, probably at Karlsruhe. There were a number of participants in that (for me) memorable discussion, and I suppose Gerhard was one of them.

Dr. Steiner brings up the very important subject of the potential of using electrical detectors as alternatives to CPCs. This view is stated in general, not just in relation to high flow DMAs, though, in our application, electrometers present the obvious advantage of readily accommodating large flow rate variations.

60

Many readers may be skeptical about electrical detectors in atmospheric measurements, given that the typical electrometers used in aerosol research have noise levels of about 1-10 fA and response times of 1 s or more. They then require over  $10^3$  elementary charges/s (or particles/s). Nevertheless, there is ample room for improving the response time and the sensitivity of most commercial aerosol

- electrometers. We recently reported the incorporation into a collecting filter of an operational amplifier 65 circuit developed by Heinz Burtscher and his colleagues, which achieved a response time below 100 ms and a noise level of about 0.1 fA (Fernandez de la Mora et al. 2017). This electrometer was commercialized by the now bankrupt company SEADM, but a comparable variant is still obtainable through my former student Luis Perez-Lorenzo. Nevertheless, this improvement is still modest
- compared to what could be achieved in practice. For instance, CCD cameras typically convert 70 individual photons into elementary charges, and then measure their current. Such rapidly evolving detectors could therefore be applied to electrical aerosol detection. That the single particle sensitivity of

CPCs is in principle extendible to electrical measurements has been clear for some time. For instance, Ma et. al. (2017) state that "... the 1.1  $\mu$ m pixel-pitch device achieves 0.21e- rms average read noise

- 75 with average dark count rate per pixel less than 0.2e-/s, and 1040 fps readout rate". Notice that the outstanding sensitivity of their CMOS-based photon-counting image sensor applies not just to a single detector, but to arrays of many detectors ordered in two dimensions. This means that DMAs with such detector arrays would not need to scan over the voltage to obtain a size spectrum. The whole spectrum would be recorded almost instantly in a multitude of detectors distributed along the inner electrode. This
- 80 possibility has already been demonstrated (Perez-Lorenzo et al. 2020) in a planar DMA with 100 operational amplifiers, though not with single charge sensitivity. Their work shows that the insulating steps separating the various metallic collectors do not cause turbulent transition. A commercial linear detector array based on CCD technology with a noise level of 0.5 fA has existed for some time (https://www.jas-sg.com/ids-2030-charged-particle-detector.html). It is 51 mm long, has 2126 active
- 85 pixels each 21 μm wide and 1500 μm high. It has been used in mass spectrometry based on instruments that disperse the ions in space. As far as we know, it has not been tested for mobility measurements, so, whether or not the collector roughness would cause turbulent transition is unknown.

An array of electrical detectors with single charge sensitivity would be clearly superior to a DMA connected to a single CPC by a large factor equal to the number of mobility channels used. Note that the

- 90 exceptional low noise achieved by Fossum and colleagues relies on the small area of each pixel. A typical situation with 1 million pixels, each 1.1 micron in length would appear at first sight to divide the signal into too many mobility channels, each receiving too little signal to be measurable in typical aerosol applications. Nevertheless, if the claimed ability by Ma et al (2017) of achieving single photon counting is correct, the signal would not be diluted over a wide mobility range, but assigned to precise
- 95 values of the mobility. High mobility resolution would not imply in this case reduced sensitivity. One could certainly add the counts from as many pixels as desired by post-processing, without loss of the available high resolution information. Of course, the fact that such developments are in principle possible does not necessarily mean that they will be available soon.

Fernandez de la Mora, L.J. Perez-Lorenzo, G. Arranz, M. Amo-Gonzalez, H. Burtscher, Fast highresolution nanoDMA measurements with a 25 ms response time electrometer, Aerosol Science and Tech., 51(6), 724 – 734, 2017

Ma, S. Masoodian, D. A. Starkey, E. R. Fossum, Photon-number-resolving megapixel image sensor at room temperature without avalanche gain, **Optica**,4 (12) 1474-1481 (2017) https://doi.org/10.1364/OPTICA.4.001474

105 J. Perez Lorenzo, R. O'Mahony, M. Amo-Gonzalez, J. Fernandez de la Mora, Instant Acquisition of High Resolution Mobility Spectra in a Differential Mobility Analyzer with 100 Independent Ion Collectors: Instrument calibration, Aerosol Sci. & Techn 54(10) 1144-1156, 2020 Anonymous referee 2

- 110 High-resolution differential mobility analyzers (DMAs) have been widely used in the sub-10 nm community over the last decade. Juan Fernandez de la Mora is completely right in framing the excellent question, why these devices have not been considered by other teams working on different applications and research questions beyond the sub-10 nm size range. I think the presented *Opinion* is therefore ideally suited to be published in such a new journal as Aerosol Research, which looks for establishing
- 115 new approaches valuable to the entire aerosol research community. While I fully enjoyed the read of that *Opinion* and I couldn't agree more with most of the excellent remarks, I forced myself to take a critical view on Juan Fernandez de la Mora's thoughts and thus would recommend publication of this *Opinion* only after a few minor points are additionally discussed in a revised version.

For the article to be called "Should high resolution differential mobility analyzers be used in
mainstream aerosol studies?", I found the discussion on why mainstream aerosol studies would benefit from higher size-resolution a bit short or maybe not highlighted well enough. I would appreciate it if at some point in the manuscript (the earlier the better) there was a list of bullet points illustrating in what sense mainstream aerosol studies would benefit from HRDMAs.

Thank you for the supporting statements.

- 125 The opinion's abstract has been extended to indicates that "... many studies central to aerosol research could be carried out with HRDMAs with considerable advantage in size range, resolution, sensitivity and measurement speed." These are the four main bullet points, right at the outset. The reason why they are not developed immediately is that there is a deeply rooted impression in the field that these advantages are either not needed or counterproductive. It seems to me it would be unwise not to give
- 130 this psychological reality full recognition from the beginning. This point is now stressed with the following two published quotations and a brief remark:

For instance, Kangasluoma et al. 2017 note that "However, these supercritical DMAs were mainly used in laboratory calibrations in previous studies because of their high flow rates of several hundred or thousand liters per minute and the corresponding high maintaining expenses". Similarly, we read in Cai et al. 2017 (2010) that

- 135 "However, these DMAs are designed to operate at high sheath flowrate (typically several hundreds to thousands lpm) rather than matching the flowrate required by the DEG-UCPC. In addition, large sheath flow recirculation system (including high volume filters and air mover) are required to operate those DMAs, making it troublesome for field measurement". These remarks must be taken most seriously, as they originate from groups working on new particle formation, which are well acquainted with the characteristics of HPDMAs.
- 140 with the characteristics of HRDMAs.

Related to the above, I find, that the current highlighting of new particle formation (NPF) and growth as the example for mainstream aerosol studies is weak. It is exactly the NPF community, which typically focuses on the sub-10 nm range, which is the size range where HRDMAs are already in use. In addition, the reference to Kong et al. (2021), which - while important - might not be the only one showing the

145 occurrence of narrow size-distributions in NPF studies (especially as it shows a highly specific case from a laboratory experiment only). The short sentences about studies on viruses and the signal-to-noise remain without reference or illustration (a Figure could nicely show this).

Excellent point. The abstract now incorporates the following remark and references:

"However, they have not been widely adopted, except in the size range below 10 nm, often in new particle formation studies (Kangasluoma et al. 2020 and Ozon et al. 2021)."

The revised manuscript now refers to published virus spectra illustrating the relation between signal/noise and resolution.

If signal processing tools are used to obtain size-distribution related parameters such as the new particle formation and growth rate, the paper by Ozon et al. (2021) investigated that a full coverage of the sizedistribution with lower resolution can be better than an insufficient coverage with high resolution (e.g. usage of a HRDMA in a DMPS with just a few mobility steps). On the other hand, Kangasluoma et al. (2020) has already highlighted in what sense sub-10 nm studies would benefit from higher resolution in the sense of uncertainty reduction. The ideal instrument would provide all together: high resolution and good coverage of all sizes at high signal (i.e. high detector flow rates). This could be clarified.

• Thank you. These two most relevant articles are now cited in the revised *Opinion*.

150

The *Opinion* (and the already posted comments) already include a very valuable discussion on higher detector flow rates, the usage of better aerosol electrometers and CPCs providing such high detection flow rates. First attempts to increase CPC detection flow rates have been made (but are still far away from "high" flow rates). It was recently shown that such an increased sensitivity through higher detector flow rates indeed provides a significantly improved performance in e.g., NPF studies (Stolzenburg et al. 2023). In addition, I just wanted to point out that for the usage of CPCs together with a HRDMA high CPC detector flow rates are no necessity for extending the Q range. q/Q can also be held constant when a CPC is used by just adding a variable make-up flow downstream of the HRDMA.

- Thank you. Stolzenburg at al. is now included in the detector discussion
- 170 I think it is important to add some thoughts on how a HRDMA system which is scanning Q would be calibrated (i.e., the set flow rates to be measured/inferred) and how fast such a system could "scan" or "step" through different flow rates. From my own experience, it seems that the blowers we currently use in our high resolution DMAs have quite a significant lag in their PID control of a new set blower flow rate and the flow to be stable and reliable. In that sense, I am skeptical if scan rates faster than typical
- 175 voltage scans can be achieved potentially contradicting the response time argument? I would appreciate it if this experimental obstacle would also be shortly discussed in the *Opinion*.

• Thank you. The following extended discussion of these important practical considerations is now included:

**• 3.9 Sheath flow rate control**

Anonymous referee 2 brings up several important issues relating to the control of the sheath gas. 180 The proposed sweeping over O rather than V requires a continuous way to measure O, which cannot be achieved with the same precision and ease as a voltage measurement. There is also the issue of the slower response time of mechanical pumps versus high voltage sources. Nevertheless, an accurate mobility scale requires a precise determination of *O*, irrespective of whether one sweeps over *Q* or over *V*. Whatever flowmeter is used, the ambiguity in the 185 mobility scale will typically be determined by the flowmeter error, which will not be necessarily greater in a *Q*-sweep than in a *V*-sweep. The main difference is that in a *V*-sweep the *Z* scale will be off by a fixed unknown factor, versus a variable equally unknown factor in a *Q*-sweep. Assuming that the relative error in the flow rate is not greater at high O than at low O, it would 190 be of little consolation to know that this error is constant. And this slight advantage will be lost in a conventional sweep if one decides not to be absolutely limited by operating always at fixed O, irrespective of the size range of interest. These considerations naturally do not remove the need to measure O as precisely as possible over the whole range of the O-sweep, which will evidently require a certain development. We are familiar with commercial flowmeters claiming 2% error in O, and reproducibility within 1%, though only covering the limited range 0-300 195 L/min. Venturi flowmeters, for instance, can go substantially higher, with comparable precision. These various flowmeters have response times much faster than the pump (limited by the inertia of the rotor), being capable of determining the actual flow rate at any instant during a sweep. The instantaneous pump frequency is also readily measured, and is an alternative marker of the 200 flow rate. There should accordingly be no difficulty in carrying out minute long full O and V scans. The possibility to achieve wide V-scans in a few seconds has been previously demonstrated (Fernandez de la Mora et al. 2017b) with electrometers, and would be far more problematic on a *Q*-sweep.

There are, however, other potential difficulties associated to a *Q*-scan that may not be so simple to handle, or that could limit the scan speed. One of them is that the working pressure in a typical closed sheath flow circuit may depend on the pump speed, while the response time for this pressure depends on how the system pressure is set relative to the external world. No experience is available at present to assess this issue.

One major experimental challenge of HRDMAs is the control of the sheath gas chemical composition, which is currently not mentioned. The *Opinion* only discusses temperature, but relative humidity and purity of the sheath gas are typically also controlled in ambient SMPS/DMPS measurements. To meet the standards which are achieved with *plain* DMAs in most ambient studies, HRDMAs would need to dry and clean the sheath flow at high flow rates. I would appreciate it if this was additionally mentioned besides the temperature issue.

- Thank you. The following extended discussion of this important practical consideration is now included.
  - The ability to control the chemical composition of the sheath gas is relatively limited, as it is unpractical to dry a large flow rate of sheath gas. The most common way of operating a HRDMA is with the sheath gas in closed circuit, such that, after an initial transient, its
- 220 composition matches that of the entering aerosol. Zinola et al. (2019) have operated a high temperature Half-Mini DMA that sampled hot automobile exhaust gases from the ambient. To avoid vapor condensation from the exhaust gases, they did run the sheath gas at 200° C. This was achieved by taking filtered ambient gas into the pump, blowing it into a heater, then into the DMA sheath gas circuit inlet, and finally exhausting the hot excess gas back to the atmosphere. The temperature was thus controlled, but not the humidity. Note that the pressure at the 225 polydisperse aerosol inlet slit in the Half-Mini DMA is close to the pressure in the separation region, which is below the pressure at the entry of the sheath gas because the flow moves at relatively high speed (Venturi effect). This pressure is also below the sheath outlet pressure because the DMA is provided with a diffuser. Accordingly, given a suitable diffuser, it is 230 possible to suck ambient aerosol when the sheath gas is exhausting to the atmosphere. On the other hand, none of the HRDMAs built to date to classify particles as large as 60 nm or more has incorporated a diffuser.
  - In the two previously tested modes, with open and closed sheath gas circuits, the only available steady compositions were either that of the ambient air or that of the aerosol. If one needs a sheath gas drier than the aerosol, one option is to dry the aerosol and operate in close circuit. Another option is to recirculate the sheath flow through a drying medium. It would be far simpler to remove the little humidity brough by the aerosol into the closed circuit than to thoroughly dry a large flow of ambient air. Other humidity controls are feasible. For instance, if the polydisperse aerosol flow is 3 L/min, and one introduces 27 L/min of dry gas into the closed circuit, the sheath gas will reach an equilibrium humidity 1/10th that of the entering aerosol.

240

235